**Subject Category:**
Biology (whole organism)

behaviour/ecology

aeroecology, flight, foraging, bat, movement ecology

**Author for correspondence:**
M. Teague O'Mara
e-mail: tomara@orn.mpg.de

# Common noctules exploit low levels of the aerosphere

M. Teague O'Mara[1,2], Martin Wikelski[1,2], Bart Kranstauber[1,3] and Dina K. N. Dechmann[1,2]

[1]Department of Migration and Immuno-Ecology, Max Planck Institute for Ornithology, Am Obstberg 1, 78315 Radolfzell, Germany
[2]Department of Biology, University of Konstanz, Universitätstrasse 10, 78464 Konstanz, Germany
[3]Department of Evolutionary Biology and Environmental Studies, University of Zurich, Winterthurerstrasse 190, Zurich 8057, Switzerland

  MTO, 0000-0002-6951-1648; DKND, 0000-0003-0043-8267

Aerial habitats present a challenge to find food across a large potential search volume, particularly for insectivorous bats that rely on echolocation calls with limited detection range and may forage at heights over 1000 m. To understand how bats use vertical space, we tracked one to five foraging flights of eight common noctules (*Nyctalus noctula*). Bats were tracked for their full foraging session ($87.27 \pm 24$ min) using high-resolution atmospheric pressure radio transmitters that allowed us to calculate height and wingbeat frequency. Bats used diverse flight strategies, but generally flew lower than 40 m, with scouting flights to 100 m and a maximum of 300 m. We found no influence of weather on height, and high-altitude ascents were not preceded by an increase in foraging effort. Wingbeat frequency was independent from climbing or descending flight, and bats skipped wingbeats or glided in 10% of all observations. Wingbeat frequency was positively related to capture mass, and wingbeat frequency was positively related to time of night, indicating an effect of load increase over a foraging bout. Overall, individuals used a wide range of airspace including altitudes that put them at increased risk from human-made structures. Further work is needed to test the context of these flight decisions, particularly as individuals migrate throughout Europe.

## 1. Introduction

The aerosphere, or the thin layer or atmosphere nearest the Earth's surface, is a dynamic habitat posing distinct energetic and behavioural challenges and opportunities to flying animals [1–4]. Animals in this habitat respond to the availability of resources (e.g. food) and environmental conditions (e.g. wind), which may be largely decoupled from the terrestrial habitat immediately below them [1]. While the aerosphere is not

uniform, open-aerial species are highly successful at exploiting it. Individuals may potentially rely on visual or acoustic cues of others moving to higher and more productive altitudes, explore higher volumes in response to low foraging success or could monitor atmospheric conditions and change behaviour based on the potential aggregation of food patches at atmospheric boundaries. While there are many possibilities of why individuals move to higher foraging altitudes, how single individuals manage to successfully forage across patchily distributed food in these large volumes, as well as their decisions to do so, are poorly understood [3,4].

Describing the three-dimensional movement of animals in the aerosphere is not only important to understand the niche they occupy, but also to delineate conservation measures. Many birds and bats are killed by human-made structures, such as power lines, or wind turbines, making accurate measurements of when, where and why they fly in the range of such installations crucial [5]. Wind turbines currently constitute one of the largest mortality risks for bats [6]. This risk is high enough that it threatens the viability of populations [7], with migratory and open-air foragers at particular risk [5,8–11]. Fatalities at turbines occur mostly when average wind speeds are lower than $5–6 \, \mathrm{m \, s^{-1}}$ [5,9–11], which also coincides with preferred wind conditions during bat spring migration [12]. The movement of insectivorous bats is particularly interesting as they appear to use a broad range of altitudes including high-altitude flights to exploit patches of migratory insects, including agricultural pests, that aggregate along nocturnal thermal inversion boundaries [13–15]. At these elevations near or above the nocturnal boundary layer, air is warmer and less turbulent, but wind speeds increase, making conditions for insect dispersal ideal [16,17]. However, it is not clear if the search for insects moves bats into higher turbine conflict zones (e.g. 30–130 m) only when winds are low, or if they regularly exploit the full altitudinal range across all wind conditions. If we are to directly assess individual risks of collision relative to remediation measures, we must have accurate measures of the heights at which bats fly.

The lack of knowledge about vertical space use is at least partly due to methodological limitations. The majority of the large existing body of data about niche division and space use, especially vertical stratification of bats, has been collected with acoustic detectors placed at relatively low distances from the ground to monitor bat activity (but see [14]). This approach cannot record individual activity, but it does provide a valuable population level overview. Detectors also have a limited detection range and their spatial coverage is limited. Questions remain as to the role altitudinal gradients play in individual foraging behaviour and how often individuals move across the aerosphere to exploit high-altitude insect migrations, which exposes them to increased mortality risk from human structures. Surveys conducted through radar and microphones connected to weather balloons or kites reflect the altitudinal potential of bat movements, but not the pattern of ascent, how long they maintain high altitudes or how this is linked to resource distribution. Most bats appear to move near the ground at altitudes lower than 30 m, but some species that show wing morphology indicative of open-aerial flight (i.e. Mexican free-tailed bats) range as high as 3000 m and their echolocation calls are regularly detected above 600 m [14,18–21].

Most bats face a morphological trade-off between the ability to fly fast and being highly manoeuvrable [22]. Bats specialized to forage in the open air have a highly adapted morphology, with small, high aspect ratio wings that are energetically efficient but come with relatively low manoeuvrability [22]. In addition, their detection range of insects is limited by the travelling distance of the returning echoes of their calls. Individual bats then have two main options to increase capture rates of prey. First, a foraging bat could fly rapidly through an air volume and repeatedly survey for prey to increase potential detection rate across multiple possible foraging patches. Alternatively, a bat could increase the tortuosity of its flight, thereby covering more of a single area in a volume-concentrated search [23]. While this may allow an individual to fully exploit a potential food patch, it would lead to lower flight speed, which decreases contact rate but increases detection rates. However, the slow flight needed to execute consecutive high-angle turns in a tortuous flight path is energetically expensive [24–26]. Likewise climbing flight is energetically demanding [27,28] with less energy recovered from descents than expended during ascent making vertically tortuous flights costly. Free-flying bats appear to forage more along a horizontal, tortuous path when in a feeding patch [29], and vertically when trying to locate a productive feeding patch [30]. Open-aerial foragers should then optimize their search for prey and minimize the amount of slowing down as well as climbing to decrease their overall foraging effort [23].

Common noctules (*Nyctalus noctula*) are ideal to test how individual open-air specialists make decisions when foraging across the aerosphere. These bats are open-air, fast fliers [30] with narrow wings and high wing loading [22,31]. They forage for a diverse range of insects [32] in short bouts

just after sunset and before sunrise. There are no sex differences in two-dimensional habitat use or the resources used when foraging [33], but there may be strong variation in foraging behaviour due to habitat and seasonal effects [34]. Common noctules integrate multiple weather cues when making migratory decisions [12], and similar integration of atmospheric conditions could dictate decisions to ascend to higher altitudes.

We asked how common noctules use the vertical space over a large lake by tracking free-flying bats with atmospheric pressure radio transmitters. While the expansion of GPS technology offers some insight into this question, GPS technology is still heavy and care must be taken when applying GPS data directly to flight height. GPS-derived heights are more prone to error than longitude–latitude data [35], especially when taken at the low temporal resolutions necessitated by the small battery sizes needed to track small bats. The technology required for radio transmission of air pressure is lightweight and eliminates the modelling needed to compensate for the altitudinal error associated with GPS measured at low sampling frequencies [36]. We predicted that bats at our study site would fly low over Lake Constance, but would show occasional high elevation foraging bouts at or near the upper limit of the nocturnal boundary layer (*ca* 200–300 m [16]) where air is warmer and less turbulent. We hypothesized that if high-elevation ascents are a response to increased foraging effort, then ascents to higher altitudes should be preceded by increased variation in altitude changes caused by the bat searching for prey vertically. Finally, we expected that bats would fly higher with increasing surface winds, which would reflect a strategy of searching higher elevations for foraging areas where insects are likely to aggregate.

# 2. Methods

## 2.1. Capture and tagging

We captured bats during the day in roost boxes in the Seeburgpark in Kreuzlingen, Thurgau, Switzerland (47.65270° N, 9.18546° E; table 1). Animals were removed from boxes and placed in soft breathable bags. We then weighed them, measured their forearm length and implanted individuals with a subcutaneous PIT tag (ID100; Euro ID, Frechen, Germany). Bats were fitted with an atmospheric pressure radio transmitter (Jim Cochran, Sparrow Systems Fisher, IL, USA) on a collar. Collars were a soft 3 mm wide shoelace [12,33,37] that we fitted to the bat's neck and secured in place with degradable braided glycolic acid suture thread (Safil-C, Braun, Aesculap). Bats tracked in 2016 had a second small transmitter (Lotek NTBQB1, 0.29 g, Biotrack, Wareham, UK) attached to the hair on their back below their shoulder blades with silicone-based skin adhesive (Sauer Hautkleber, Manfred Sauer, Germany) to monitor migration timing. The average mass of a bat was 28.6 ± 0.13 g (mean ± s.d., range 23–33 g). Total transmitter package weight was less than 1.3 g (4.47 ± 0.55% body mass ± s.d., range: 3.9–5.43%). Transmitters had no measurable short-term impact on bat body mass or behaviour. One bat (165.068) was recaptured immediately after foraging and had gained 8 g over previous capture mass (27%). He had continued to gain mass when he was recaptured 24 days later.

## 2.2. Extracting flight altitudes and wingbeat data

The atmospheric transmitters [38] emitted a continuous long-wave signal that was recorded from locations on the ground using wide-range radio telemetry receivers (AR8000/8200, AOR Ltd; Sika, Biotrack) with collapsible H- or Yagi-antennas and then recorded to a sound file (44.1–48 kHz) on a digital recorder (Tascam DR-05) via mini-dv output. We also tracked bats in the air from a Cessna 172 airplane [12,33]. The detection distance was 2–3 km from the ground and *ca* 8 km from the airplane. We did not triangulate longitude–latitude position due to logistical constraints, but did note foraging location (areas of intense, repeated use) based on the radio signal strength and direction. The radio signal included a 24-bit digital transmission that coded air temperature and air pressure (to 0.01 mbar or *ca* 0.1 m) as well as a 3 s beacon (figure 1). These transmitters therefore give air pressure information every 6 s. The data transmission (figure 1, 0–2.5 s) is followed by a constant beacon (2.5–6 s) where the amplitude of the signal is modulated by the vertical movement of the antenna as a consequence of the bat's wingbeat. This movement modulates the amplitude of the received radio signal, and wingbeats can be observed (figure 1a, asterisks). The altitude transmitted by transmitters had been previously calibrated for accuracy using the altimeter of a small aircraft with $r^2 = 0.99$ [38]. We manually counted the wingbeats of flying bats in a subset of the recordings using the amplitude

**4**

**Table 1.** Daily sample sizes (*N*), flight heights, cumulative height change and wingbeats for each bat. Medians are given with the interquartile range (IQR).

| bat ID | date [*N*] | sex | mass (g) | tag Pct mass | forearm length (mm) | max height (m) | median height (IQR) | cumulative height change (m) | median wingbeat frequency (IQR) [*N*] |
|---|---|---|---|---|---|---|---|---|---|
| 150.348 | | F | 25 | 5.00 | 54.4 | | | | |
| | 7 May 2016 [481] | | | | | 327 | 23.8 (18 – 30.6) | 1868 | 7 (6.2 – 7.5) [481] |
| | 8 May 2016 [341] | | | | | 38 | 17.2 (13.8 – 21.4) | 638 | 7.7 (7.3 – 7.7) [341] |
| | 9 May 2016 [213] | | | | | 30 | 15.6 (13.5 – 17.6) | 395 | 7.3 (7 – 8) [213] |
| 150.445 | | F | 23 | 5.43 | 51.1 | | | | |
| | 4 May 2016 [286] | | | | | 167 | 27.1 (23.8 – 31.8) | 1382 | 7.5 (7 – 7.9) [286] |
| | 8 May 2016 [267] | | | | | 67 | 17.1 (11.8 – 52.2) | 567 | 6.9 (5.5 – 7.5) [267] |
| | 9 May 2016 [86] | | | | | 119 | 13.6 (9.6 – 89.1) | 708 | 8 (6 – 10) [86] |
| 150.665 | | F | 30 | 4.16 | 55.6 | | | | |
| | 4 May 2016 [219] | | | | | 123 | 21.4 (12.2 – 30.5) | 740 | 7.8 (7 – 8.7) [219] |
| | 5 May 2016 [111] | | | | | 139 | 43.1 (26.7 – 75.9) | 546 | 8.2 (7.5 – 13.8) [111] |
| 164.313 | | F | 26.5 | 4.15 | 53.0 | | | | |
| | 3 May 2017 [235] | | | | | 58 | 25.8 (1.9 – 31.4) | 553 | 7 (6.7 – 8) [235] |
| 165.200 | | M | 33 | 3.94 | 53.7 | | | | |
| | 11 Aug 2014 [376] | | | | | 26 | 7.1 (6.5 – 11.1) | 426 | 7.4 (5 – 8.1) [376] |
| | 12 Aug 2014 [141] | | | | | 81 | 17.3 (16.4 – 17.9) | 423 | 7.6 (7 – 8) [141] |
| | 13 Aug 2014 [111] | | | | | 89 | 22.4 (15.7 – 32.6) | 654 | 8.4 (7.5 – 9) [111] |
| | 14 Aug 2014 [28] | | | | | 130 | 59.1 (24.8 – 73.7) | 437 | 7.7 (7 – 8.9) [28] |
| | 15 Aug 2014 [20] | | | | | 29 | 19.3 (16.4 – 25.9) | 76 | 7.6 (7.2 – 8.6) [20] |
| 165.135 | | M | 31.5 | 4.12 | 54.0 | | | | |
| | 1 Sep 2014 [103] | | | | | 30 | 17.1 (14 – 21) | 299 | 8.2 (7.6 – 9.4) [103] |
| | 3 Sep 2014 [102] | | | | | 90 | 6.4 (6.1 – 6.7) | 252 | 7.5 (7.2 – 7.9) [102] |
| | 5 Sep 2014 [178] | | | | | 117 | 28.3 (19.8 – 37) | 1084 | 7.7 (7 – 8.5) [178] |

(*Continued.*)

**Table 1.** (*Continued.*)

| bat ID | date [N] | sex | mass (g) | tag Pct mass | forearm length (mm) | max height (m) | median height (IQR) | cumulative height change (m) | median wingbeat frequency (IQR) [N] |
|---|---|---|---|---|---|---|---|---|---|
| 165.068 | | M | 32 | 4.06 | 53.8 | | | | |
| | 18 Aug 2014 [337] | | | | | 67 | 22.9 (17.2−35) | 1284 | 8.3 (7.7−9.1) [337] |
| | 19 Aug 2014 [360] | | | | | 52 | 16.9 (12.8−22.6) | 1386 | 8 (7.3−8.5) [360] |
| | 20 Aug 2014 [294] | | | | | 65 | 30.6 (25.6−36.7) | 906 | 7.5 (6.9−8) [294] |
| | 15 Sep 2014 [43] | | 34 | 3.82 | | 55 | 30.2 (24.3−42.9) | 234 | 7 (6−7.9) [43] |
| | 16 Sep 2014 [152] | | | | | 67 | 24.8 (18−30.8) | 366 | 7 (6.7−7.5) [152] |
| 165.183 | | M | 29 | 4.48 | 54.6 | | | | |
| | 13 Sep 2014 [16] | | | | | 9 | 7.9 (7.6−8.4) | 9 | 7.6 (7.4−7.7) [16] |
| | 14 Sep 2014 [46] | | | | | 120 | 3.1 (12.6−63.8) | 315 | 7.9 (6.9−8.2) [46] |

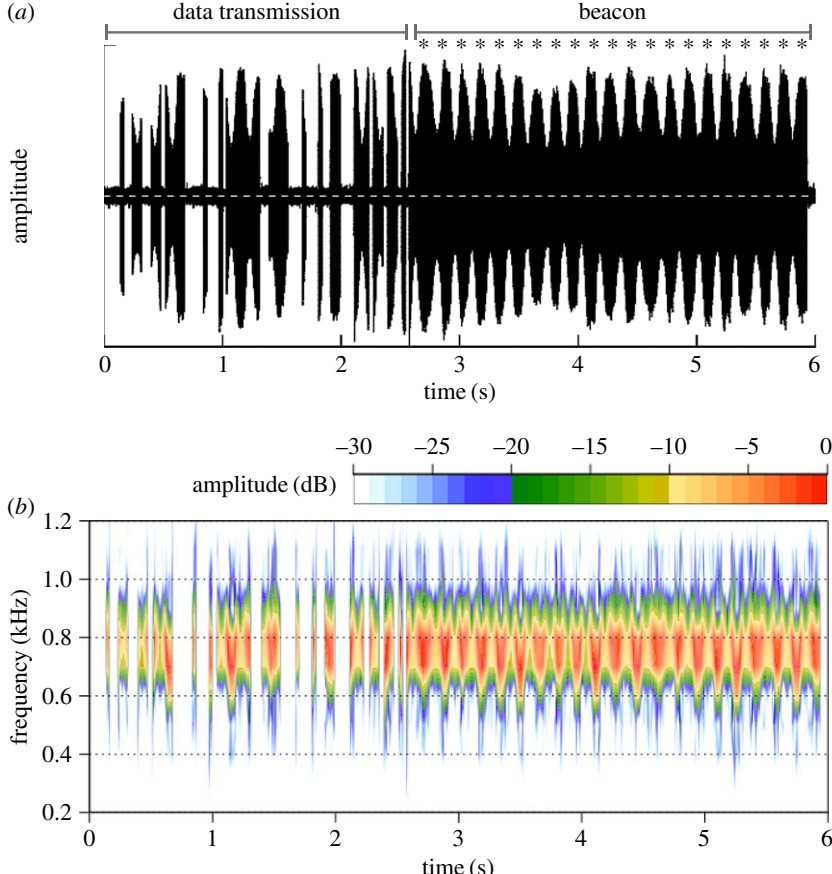

**Figure 1.** Raw data signal in (*a*) waveform and (*b*) spectrogram from the atmospheric radio transmitters of a bat in flight used to calculate flight altitude and wingbeat frequency. The 24-bit data transmission (0 – 2.5 s) is followed by a constant beacon (2.5 – 6 s) where the signal amplitude is modulated by the vertical movement of the antenna. This movement modulates the amplitude of the received radio signal. Wingbeats can be observed at the asterisks.

modulation of the beacon (figure 1*a*, asterisks) caused by the displacement of the radio antenna with each wingbeat [39–42]. Recording quality was variable and was strongly autocorrelated within a recording. We extracted data only from recordings where a minimum of 1 s of the beacon signal showed clear amplitude density variation, so that the presence or absence of wingbeats was clear.

To decode the recorded radio signal, we employed an iterative process using *tuneR* [43] and *seewave* [44] in R 3.5.1 [45] to identify the timing of the bit position from each data transmission. First, the plausible start of the data-coding period was identified by finding the maximal difference between periods that are known to be transmitting and silent. From these plausible starts, a template of the signal was then used to identify a selection of audio of the same length as the data pattern. The selected audio was filtered to a range between 300 and 10 000 Hz. The signal was then smoothed using a window length of 180 samples and an overlap of 98%. This was followed by finding an amplitude cut-off that identified the changes in the on/off position of the bits which aligned with the template. We also accounted for slight variations in signal speed and variation in start time. We then checked the resulting classification by comparing the known silent and signalling periods with the classification. We updated the procedure based on interpolating from other identified start times using three iterations. We removed measurements where the start bit could not be identified, bits were missing or which yielded questionable temperature or pressure readings. This reduced the original sample size by 13.0% (from 6472 to 5627) that then had all locations where bats were roosting removed for a final sample size of 4546 air pressure measurements, with a median time lag of 6.0 s (mean $11.8 \pm 11.2$ s) between consecutive air pressure observations. We converted the decoded air pressure (mbar) to height above sea level using normal temperature and pressure at sea level, and then subtracted the elevation of Lake Constance (395 m.a.s.l.). All data are available at the Movebank Data Repository (doi:10.5441/001/1.7t4b97qf [46]), and weather-annotated data are available from the Dryad Digital Repository (doi:10.5061/dryad.63q3283 [47]).

## 2.3. Weather data

Hourly weather data for air pressure, air temperature and meteorological wind direction were downloaded from the German Weather Service (werdis.dwd.de) for the Konstanz weather station (47.67742° N, 9.190052° E, 442 m.a.s.l.), 2.7 km away from our capture site. We used a time-weighted interpolation to match our observations to the changing weather conditions.

## 2.4. High-altitude ascent and foraging effort

To test if higher-elevation trips were related to poor foraging conditions and increased foraging effort, we first identified ascents where bats climbed continuously for more than 40 m ($n = 7$). For example, figure 2, bat 150.348 shows an individual with large variation in height early in the night before it makes a rapid high ascent, after which there are relatively small changes in height. As ascending flight incurs high energetic costs and bats seem to use more horizontal movements when in a feeding patch [29], and move vertically when foraging [30], we assume that flights with large changes in height indicate greater foraging effort. We therefore calculated the coefficient of variation in the difference in height between consecutive observations for 2, 5 and 10 min before each ascent (pre-ascent). We also calculated the coefficient of variation in height difference for all consecutive observations in the dataset at the same time intervals. We compared the observed pre-ascent coefficient of variation to the full distribution to identify if the periods before higher ascents had more (greater than 95% quantile) or less (lower than 5% quantile) variation than all observations.

## 2.5. Analysis

All analyses were conducted in R 3.5.1 using a combination of linear models, non-parametric statistics and linear mixed effects models in *lme4* [48] with $R^2$ calculated in *MuMIn* [49,50] where appropriate. We also employed a custom permutation test to test for mean differences in ascending and descending behaviours. This first pooled all observation and then randomly drew the same number of observations for each defined starting group. The differences in means were then calculated and compared to the mean difference between the original groups. R Code is provided in the electronic supplementary material.

# 3. Results

## 3.1. Flight altitudes

We tracked eight bats during one to five foraging sessions each for a total of 25 sessions (23 evenings, two mornings). Bats emerged from their roost boxes $21.56 \pm 7.40$ min after sunset and flew for $87.27 \pm 24$ min before returning to roost. We recorded the bats' general location over Lake Constance and the area around it based on direction and signal intensity and found that similar to a previous study bats foraged primarily at the margins of the lake and up the Rhine river to a large wetland and reed bed areas [33]. We recorded 4546 height locations during these foraging sessions. Bats flew $25.3 \pm 22.3$ m above Lake Constance, covering 9–1868 m of cumulative height change per night (table 1, mean: $647 \pm 459$ m), and this total height is strongly correlated with number of locations recorded (Pearson's correlation = 0.76, $t_{22} = 5.49$, $p < 0.001$). While bats predominantly flew at low altitudes, they ascended to over 100 m during 16 of 25 observation sessions and each bat used a variety of strategies throughout foraging (figure 2). Some chose to immediately explore over 100 m (figure 2, bat 150.665), others took brief trips to higher elevations with multiple ascents over 40 m (figure 2, bat 150.348) while others always remained low over the lake.

Most flights to heights over 100 m were short with a rapid ascent and descent. In only two flights did bats spend more than 1 min (ca 5 min) at a relatively stable height above 100 m. The mean foraging heights varied considerably across bats and nights (table 1). Across all flight segments, bats descend faster than they ascend (median, IQR: descent: 0.14 m s$^{-1}$, 0.06–0.34; ascent: 0.15 m s$^{-1}$, 0.06–0.34; permutation test: mean difference = $-0.137$, $N_{\text{Ascend}} = 1622$, $N_{\text{Descend}} = 1543$, $p = 0.011$; figure 3a), and the fastest rates were found on descent with time lags between height fixes of less than 10 s (Pearson correlation: $-0.10$, $t_{3163} = -5.90$, $p < 0.001$). However, when bats executed flights with over 40 m in continuous height change, they ascended faster than they descended (median, IQR: ascent: 0.33 m s$^{-1}$,

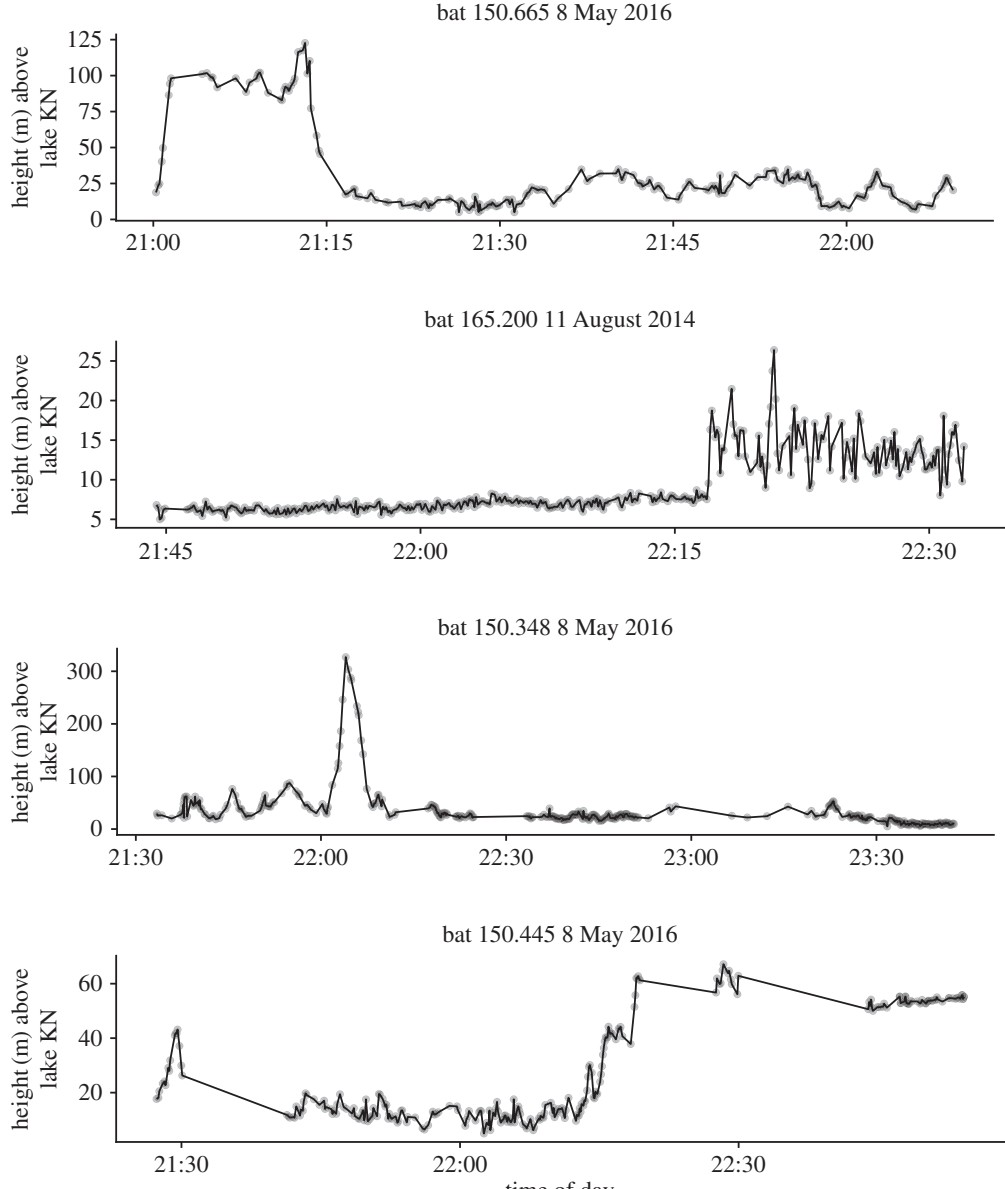

**Figure 2.** Example height profiles of four noctules. Noctules foraged mostly at low altitudes but included flights up to 300 m above Lake Constance. Note that axes are not equal across panels.

0.25–0.96; descent: 0.12 m s$^{-1}$, 0.08–0.22; permutation test: mean difference = 0.600, $N_{\text{Ascend}} = 7$ $N_{\text{Descend}} = 11$, $p = 0.0495$; figure 3$b$).

## 3.2. Weather influence on flight height

There was no explanatory relationship between the heights to which bats flew and wind speed, air temperature, wind direction or air pressure (electronic supplementary material, figure S1). However, there was relatively little variation in our weather variables within each year (electronic supplementary material, figure S2). Wind speeds were mostly less than 3 m s$^{-1}$, and air pressure changed by only a few millibar within each year (electronic supplementary material, figure S2). Air temperature varied by 2–10°C within a year, but this did not have any measurable effect on bat foraging heights.

## 3.3. High-altitude ascent and foraging effort

We did not find that larger ascents were preceded by increased or decreased variance indicative of decreasing foraging success regardless of whether we included 2, 5 or 10 min before the ascent

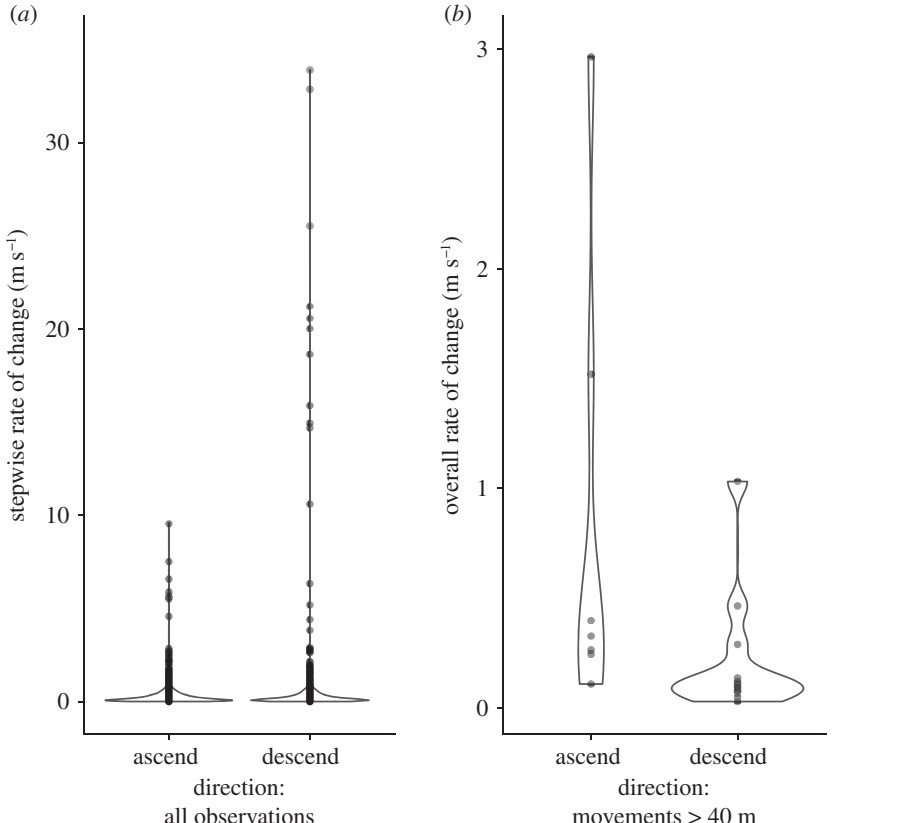

**Figure 3.** Ascent and descent rates of (*a*) all stepwise changes in height and (*b*) overall rates of change for complete flights with over 40 m of height change. Bats descended faster than they ascended across all stepwise observation (permutation test $p = 0.011$), but bats ascend faster than they descend on flights over 40 m (permutation test $p = 0.0495$).

relative to the full distribution of all observations (figure 4). There was little correlation between the total height of the ascent and the pre-flight variance (Kendall's $\tau = 0.38$).

## 3.4. Wingbeats

We recorded 2376 wingbeat frequencies subsampled from all eight bats over several nights. Wingbeat frequencies ranged from 0 to 15 beats per second (bps), with an overall median of 7.5 bps across all individuals. The individual median wingbeat frequency was positively, but moderately related with capture mass (wingbeat frequency (bps) = 0.0698 × mass (g) + 5.554; $F_{1,6} = 6.30$, $p = 0.046$, $R^2 = 0.51$, figure 5), but we did not detect a relationship between wingbeat frequency and size (forearm length). We then tested if the time of night explained wingbeat frequency because bats can change mass by up to 30% during nightly foraging. We created a linear mixed effects model to assess the influence of time of night on wingbeat frequency. To do so, we limited the data to the first foraging session of the night and included individual identity as a random effect. We then used a likelihood ratio test to assess the effect of time of night on the model and found that bats increase their wingbeat frequency across the night ($\chi_1^2 = 6.302$, $p = 0.012$; slope = $0.178 \pm 0.07$ s.e.m.). This relationship has weak explanatory power ($R^2_{\text{marginal}} = 0.005$) that is increased when individual identity is accounted for ($R^2_{\text{conditional}} = 0.10$). High-elevation ascent and descent flights did not differ in their median wingbeat frequency (permutation test: mean difference = 0.025, $N_{\text{Ascend}} = 7$, $N_{\text{Descend}} = 11$, $p = 0.959$). Noctules skipped wingbeats or glided in 9.9% (236 of 2376) of our observations. Gliding occurred more frequently at higher altitudes ($26.9 \pm 16.36$ versus $30.4 \pm 16.20$ m; permutation test: mean difference = $-6.04$ m, $N_{\text{Ascend}} = 2140$, $N_{\text{Descend}} = 236$, $p < 0.002$), but did not occur more often on descent than ascent or level flight ($\chi_1^2 = 0.04$, $p = 0.84$).

## 4. Discussion

Unlike other narrow-winged bats that regularly exploit the atmospheric boundary layer, common noctules seldom foraged at high altitudes. Instead, we found that over Lake Constance, they foraged

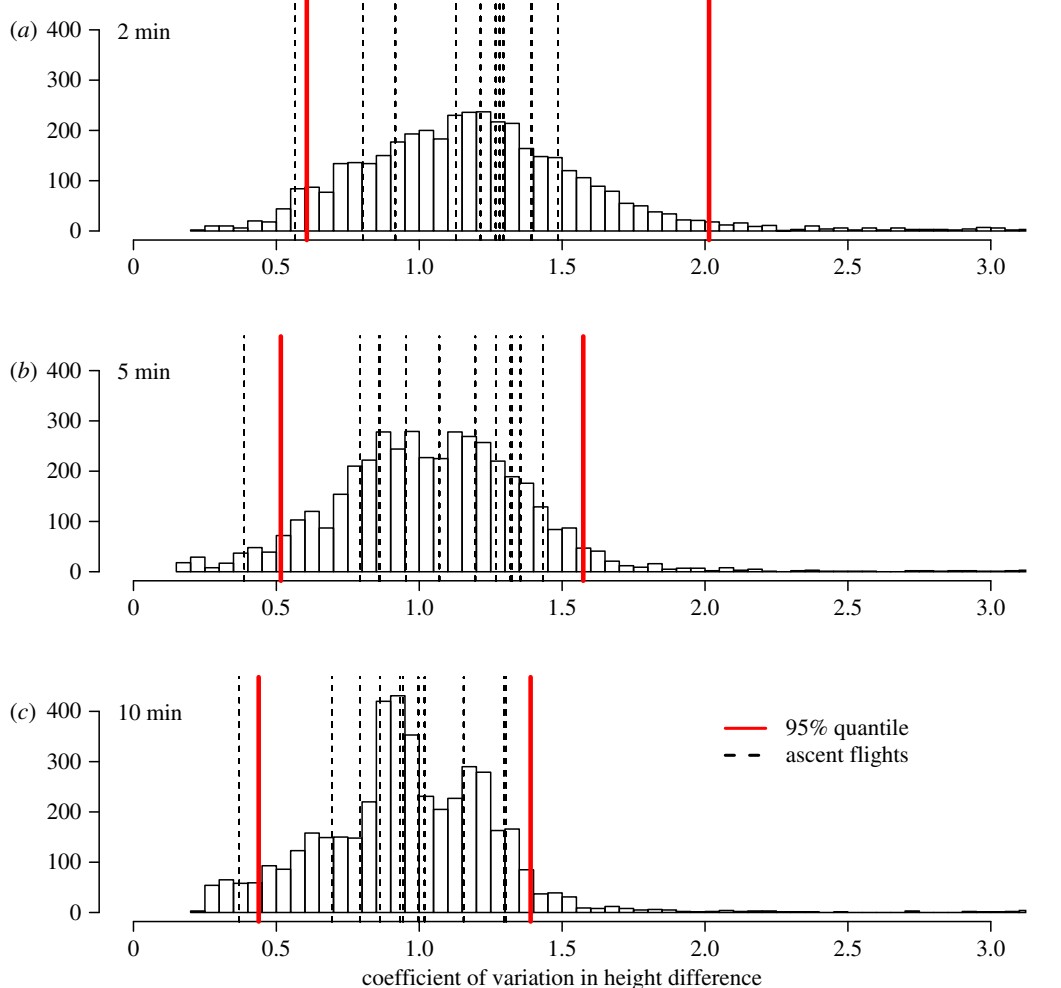

**Figure 4.** Coefficient of variation in differences in consecutive height measures prior to individual ascents over 40 m. Individual flights are marked by dashed lines, and the full distribution of the coefficient of variation in the height differences for (*a*) 2, (*b*) 5 and (*c*) 10 min intervals across all observations are shown. The 95% quantile of each distribution is marked in red.

at relatively low altitudes with only occasional short exploratory flights to 100–300 m above lake level. Against expectations, flight altitude was not influenced by weather variables measured at hourly intervals, and ascents to higher altitudes did not follow an increase in searching across altitudes, which would have indicated increased overall foraging effort. In general, noctules descend faster than they ascend, but when moving over 40 m in elevation, they climb faster than they descend. Noctules modulated wingbeat frequency over a large range while in flight, but this was not related to altitude changes. Surprisingly, they also regularly skipped several wingbeats in short gliding bouts.

The low general flight heights and few higher ascents indicate that sufficient insects were distributed within a few metres of the lake's surface. Common noctules are opportunistic foragers that feed on a wide range of insects, including water emergent swarms of Trichoptera and Diptera, as well as Hemiptera, Lepidoptera and Coleoptera when swarming insects are not available [32,51]. During our observations, as well as during past observations at this site, noctules foraged over the lake margin, reed beds and low-lying surrounding areas [33]. The low altitude foraging strategy we observed is consistent with previous work that has estimated flight altitudes of noctules using other techniques with varying precision. Radar tracking found that noctules migrating over the Baltic sea fly lower than 10 m [52], but may ascend to over 1500 m [53]. Based on traditional radio tracking, Kronwitter [54] suggested that noctules forage up to at least 250–500 m because of exceptionally clear radio signal reception of over 8 km. Over mixed terrestrial landscape and recorded with GPS at 30 s intervals, noctules generally flew at or below 50 m and, with a more limited number of locations, up to 250 m [55].

It is unclear why our bats chose to ascend hundreds of metres for only short periods of time, particularly if foraging success was high at low heights. Several narrow-winged open-aerial bat

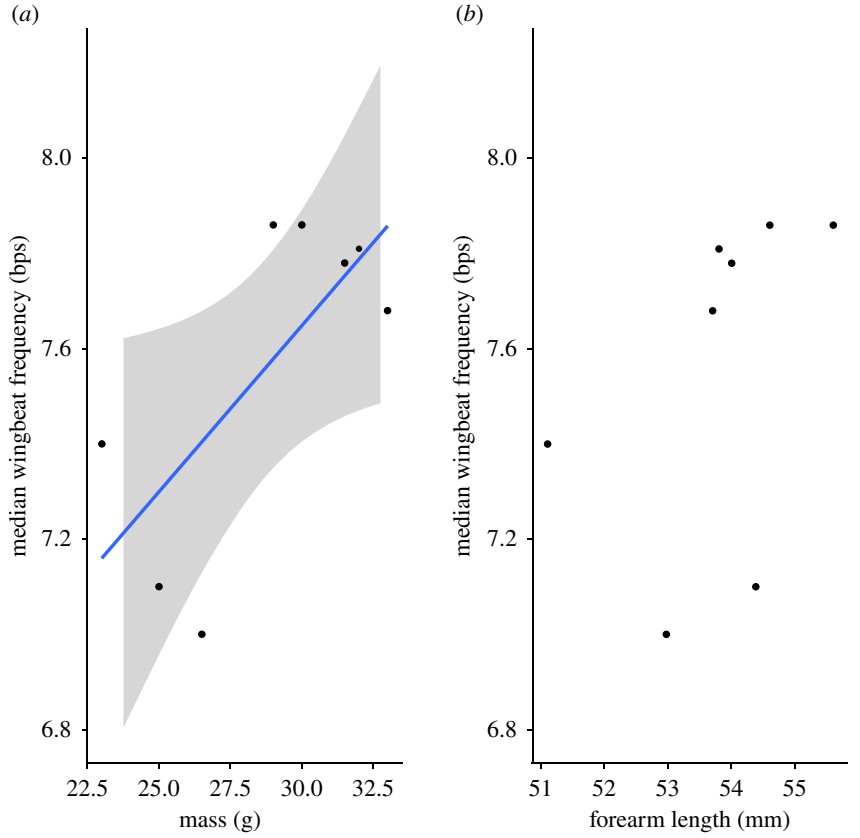

**Figure 5.** Median wingbeat frequency (bps) for each individual bat and its (*a*) capture mass and (*b*) forearm length. Capture mass was positively related with median wingbeat frequency (wingbeat frequency (bps) = 0.0698 × mass (g) + 5.554, $R^2 = 0.51$). The 95% confidence interval is shown in grey.

species forage at altitudes of over 1000 m.a.s.l. for insects [14,18,56–60], but high-altitude flight still remains a rare aspect of foraging behaviour. It is difficult to determine from acoustic surveys if individuals remain at these altitudes to forage, or if the high-altitude samples result from short ascents either for exploration or prey capture similar to our observations with the noctules. Unfortunately, we have neither acoustic nor diet information to assess if these higher altitude flights were foraging movements, but the rapid exploratory ascent and descent were similar to altitudinal foraging of purple martins (*Progne subis*), a diurnal open-air specialist. Purple martins sampled with accelerometers every 20–30 s ascend rapidly to areas with increased insect density, capture prey and then descend immediately to feed nestlings [61,62]. By contrast, common swifts (*Apus apus*) provisioning nestlings tend to stay at the same altitude and move in a consistent general direction as they fly through a volume [23]. Our bats mostly stayed at level flight but were able to move over 30 m up or down in the 6 s or less between fixes. This is consistent with anecdotal reports of common noctules consistently ascending and descending across a small range of heights until a food patch is encountered [30].

The predominantly low flight altitudes also show that the physical dynamics of the aerosphere appeared to have little influence on the altitudinal selections made by the noctules that we tracked. The lack of association between weather variables, particularly air pressure and wind speed, and high-elevation flights suggests that the near-surface lake environment is highly productive, and that noctules could easily meet foraging needs in the 80 min that bats were outside of their roosts during the observation periods. Furthermore, there was relatively little variation in the weather variables within each tracking season. The exception is air temperature in 2016 that spanned 8°C, but these nocturnal conditions at Lake Constance might not facilitate a temperature inversion that is more conducive for insects to aggregate aloft at the upper limit of atmospheric boundaries [13,15,63–65]. This nocturnal boundary layer with a decrease in air turbulence, increase in air temperature and increased wind speeds can be highly mobile across the night [16,17], but generally extends hundreds of metres above the surface. Finally, we find these large ascents from both females in the spring

migrations season and from males that are resident in the autumn. If noctules only ascend to make larger movement decisions, we would expect that these ascents would be more common from females that are deciding to migrate towards maternity colonies or overwintering sites.

Weather variables, especially wind, could also have an important influence on wingbeat frequency, reflecting the effort bats have to invest into maintaining their speed and altitude. Our noctules used a range of wingbeat frequencies with a median frequency that was slightly faster than noctules flying in a wind tunnel at a steady rate of $6 \, m \, s^{-1}$ [66]. However, to understand the relationship between wingbeat frequency and weather variables such as wind speed, the speed and direction of travel is needed to make reasonable inferences about how bats modify their flight behaviour. We did not record longitude–latitude location of the bats and cannot make a reasonable inference as to travel direction and speed relative to wind direction; therefore, we only generally tested the influence of weather variables on wingbeat frequency.

Bats did not change their wingbeat frequency as they made ascents and descents over 40 m, which indicates that like both pigeons [27] and lesser dog-faced fruit bats (*Cynopterus brachyotis* [28]), the aerodynamic power needed for climbing ascents comes through other adjustments of wingbeat kinematics. We did find a positive relationship between the mean wingbeat frequency and capture mass, which is consistent with experimental work testing the effects of increased load on horizontal [67] and ascending flight [28]. Bats in these captive studies increased wingbeat frequency rather than wingbeat amplitude to increase total power generation, especially to support the vertical aspect of their trajectory. Bats in our study showed a wide range of wingbeat frequency within a single night, and there was a general, but weak, increase in wingbeat frequency as the night passed and bats returned to their roost nearly 30% heavier after foraging.

Radar and video snapshots of free-flying noctules show airspeeds near their power optimum [66] of $6.0 \pm 2.1 \, m \, s^{-1}$, and that they beat their wings at 7.0 to $7.5 \pm 0.63$ bps (5.5–8 bps) with periodic glides that last 0.5–0.7 s [30,68]. Our longitudinal tracking of individual common noctules showed pauses in wingbeat frequencies similar to Mexican free-tailed bats [42], that can last over an entire second. Bouts of wingbeat pauses were more common in free-tailed bats as they flew faster [42], and it is likely that the same is true for common noctules. However, unlike captive pipistrelle bats that pause for one to two wingbeats during quick descents [69], wingbeat pauses were more common across all flight directions. Further tracking that resolves three-dimensional space use as well as wingbeat kinematics would be necessary to understand how and when common noctules glide.

Bat flight heights are impacted by the landscapes over which they fly and their behavioural mode (e.g. foraging versus migrating), which can lead to challenges in setting conservation priorities across the year. Noctules are the most common species killed at wind turbines in Germany [5]. While mortality risk for noctules and other migratory bat species is higher during their migration flights, turbines kill individuals from both local and migratory populations [70]. Mortality risk for bats increases with turbine height and rotor diameter [5], and bats tend to visually orient towards turbines [11]. The broad range of individuals killed by turbines reflects the diverse heights at which bats fly, and our data suggest that even though noctules may not spend significant time at collision-related heights, all individuals crossed the turbine danger zone of 40–150 m. Clearly, there is no simple general picture of how bats use the aerosphere, and better data on the seasonal and geographical variation in flight behaviour including altitude are warranted to determine the time, conditions and populations that put bats at higher collision risk.

Ethics. Handling and sampling of the bats was approved by the Veterinäramt Thurgau (FIBL1/12) and all methods conformed to the ASAB/ABS Guidelines for the Use of Animals in Research.

Data accessibility. All height and wingbeat data are available from on Movebank (movebank.org, study name: 'Foraging heights of common noctules (data from [46])' and are available at the Movebank Data Repository (doi:10.5441/001/1. 7t4b97qf), and weather-annotated data are available from the Dryad Digital Repository (doi:10.5061/dryad.63q3283).

Authors' contributions. M.T.O., D.K.N.D. and M.W. conceived and carried out the experiments. M.T.O. and B.K. analysed the data and contributed analytical tools, M.T.O. wrote the first draft of the manuscript and all authors contributed to the revisions and approved the submitted version of the manuscript.

Competing interests. The authors declare no competing interests.

Funding. Funding for this project was provided by the Max Planck Institute for Ornithology and the University of Konstanz.

Acknowledgements. This work was made possible by the University of Konstanz Vertiefungskurs 'Biology Going Wild' students who tracked bats in 2016 and 2017. We would also like to thank Jennifer and Selim Golbol, Anne Scharf, Mariëlle van Toor, Monika Rauser, Monika Krome, Julian Wanner, Marion Muturi, Alaa Eldin Soultan, Inge

Müller, Brigitta Keeves and Kate Ihle for their help with tracking. Marion Muturi assisted with wingbeat counts, and Kamran Safi provided additional analytical and logistical support. We would also like to thank Peter Bergsteiner from the Stadtverwaltung Kreuzlingen and the Seeburgpark staff for access to the park. We would like to thank Brock Fenton and Gerry Cater for their suggestions that improved this manuscript, and Gerry Carter for providing the code for the permutation test.

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
