## [Reviewer comments · Royal Society Open Science]

Review History

RSOS-181942.R0 (Original submission)

Review form: Reviewer 1 (Brock Fenton)

Is the manuscript scientifically sound in its present form?

Yes

Are the interpretations and conclusions justified by the results?

Yes

Is the language acceptable?

Yes

Is it clear how to access all supporting data?

Yes

Do you have any ethical concerns with this paper?

No

Have you any concerns about statistical analyses in this paper?

No

Recommendation?

Accept with minor revision (please list in comments)

Comments to the Author(s)

I salute you all on such an excellent contribution. The topic is so alluring and yet has received so little attention. Thank you for sorting this out.

I do, of course, have a couple of suggestions.

First, please put more information into the abstract ...give the numbers of bats tagged (8) and the time spent in contact with them this will put the size of the data base in clear context and it is appropriate to make it easier for the reader to get an idea of what is to come.

Second. Consider articulating the hypothesis that guided the work. I like the specific predictions, but if they arose from a guiding hypothesis, the impact of the work would be clearer to everyone. This will increase the impact and readership of the contribution.

Great work, lovely contribution

Brock Fenton

Review form: Reviewer 2 (Gerald Carter)

Is the manuscript scientifically sound in its present form?

Yes

Are the interpretations and conclusions justified by the results?

Yes

Is the language acceptable?

Yes

Is it clear how to access all supporting data?

Yes

Do you have any ethical concerns with this paper?

No

Have you any concerns about statistical analyses in this paper?

Yes

Recommendation?

Accept with minor revision (please list in comments)

Comments to the Author(s)

This paper describes the altitudinal movements of common noctule bats using pressure-sensitive radio-transmitters. The study is very well-written, and it provides novel and valuable information about how bats use the vertical dimension. The bats ascended briefly over 100 m in most foraging sessions, but they stayed >1 minute above that height in only 2 of 25 observed foraging sessions. The results made sense given what we know about how these bats forage. The authors did not detect that the bats increased their wingbeats during these ascents, but the bats did skip wingbeats and also glided. This information could be useful for informing policies for

mitigating bat mortality at wind turbines. It is also terrific to see what animals are doing in places where we cannot normally observe them.

There were just a couple of places that I think the manuscript could be improved. First, it would be nice to get a bit more detail of the accuracy of the tags (given the possible change of air pressure caused by wingbeats). I suspect this information is somewhere and it would just be nice to provide it for the reader since so much depends on a rough idea of their accuracy. One way to do this (maybe) would be to put some standard-size error bars on the altitude data so one could infer what is a bat moving up and down versus what is simply error. Or if the tags are high-resolution enough, perhaps there would be no visible error bars?

Second, it would help to draw a better distinction between results of “no relationship” that the authors think actually means a relationship of zero (biologically irrelevant) versus when “no relationship” means something like “we detected no relationship probably due to a small effect size and a lot of noise”. My line comments for each section are below. For more powerful nonparametric tests, I provide some R code for an alternative to rank tests.

INTRO

First paragraph: Regarding the “aerosphere” niche, can you briefly clarify what you mean using some concrete examples? Is an insect flying one foot above the ground in “airspace” or are we only talking about very high up? (e.g. birds and bat feeding on migrating insects/birds/bats). What is the difference between airspace and aerosphere? How high does it need to be?

50-53 Sentence needs revision.

69 Define “higher conflict zones”

77 missing parenthesis

126 bats at our study site

129 A bit confusing: the first part is the hypothesis, and the second part is the prediction? Maybe reword as:

“If high-elevation ascents are a response to increased foraging effort, then ascents to higher altitudes should be preceded by increased variation in altitude changes caused by the bat searching for prey vertically.”

METHODS AND RESULTS

161 Can you briefly clarify how you measured the general foraging location? (e.g. using radio signal strength to get distance and direction?)

168 “The transmitters had been previously calibrated for accuracy using the altimeter of a small aircraft with $r^2 = 0.99$ [38].” Can you give more information here? What is the scale or unit of analysis for $r^2=0.99$? Given the wingbeats affect on air pressure, what is the error bars (roughly speaking) that are associated with altitude measures for a flying bat? For example, are these tags accurate to 0.1 meter, 1 meter, 10 meter, 40 meters?

197 Sample size is 4546 air pressure measures?

Figure 1. This plot is your method for counting wingbeats? If so, maybe say that in Figure legend.

219 “we assume that flights with large changes in height indicate greater foraging effort.” Does dropping in altitude also indicate foraging?

Figure 2. I think it would help to connect the dots with a line so we can see the variation over time (or the error) more clearly, and to expand the x-axis and stack the plots in 4 rows. Figure 2 might be a good place to illustrate how you interpret the data. How many ascents are counted in each panel?

Fig. 3A says “there are no differences” but the distributions looks different. By ranking the data, you cannot detect that the high values are much higher in the descending rates, because the exact values are lost. Data look lognormal, and might be better to visualize and analyze them after a log transformation like $\log(x+1)$.

A nonsignificant test doesn’t show that two variables are similar or drawn from the same distribution. So I think it’s better to say “no differences detected” rather than saying “the same”, “no effect”, “no differences” or “no relationship” when not detecting a difference with low power (unless the mean effect is precisely near zero). This also applied to lines 294, 302, or 305 and first paragraph of discussion.

Many readers do know when to translate “no effect” as “inconclusive”, but many do not. If you do think the effect is about zero, I think it makes sense to say “no effect” but then you should present a near-zero effect size and confidence interval that is close around zero.

Also, I think ranked data tests like Mann-Whitney U tests are just too low power to argue convincingly for differences or lack of differences between two groups. Below is some R code for a more powerful non-parametric permutation test that can be used of Mann-Whitney instead to test differences between two means drawn from any distribution.

I think the Figure 4 caption (line 285 or 286) is missing some words (e.g. “are shown”)

Besides my comment above about “no relationship” vs “not detecting a relationship” , the discussion reads well to me.

```
-----
# PERMUTATION TEST-----
permutation.test <- function(x,y){
  perms <- 5000 # get number of permutations
  n1 <- length(na.omit(x)) # get sample size of group 1
  n2 <- length(na.omit(y)) # get sample size of group 2
  obs <- mean(x, na.rm=T) - mean(y, na.rm=T) # get mean difference
  exp <- rep(NA, perms)
  for (i in 1:perms) {
    # shuffle data to random groups
    xy <- sample(c(x,y))
    x2 <- xy[1:n1]
    y2 <- xy[(n1+1):length(xy)]
    #get mean difference between random groups
    exp[i] <- mean(x2, na.rm=T) - mean(y2, na.rm=T)
  }
  # get two-sided p-value
  p <- mean(abs(obs)<=abs(exp))
  # get results
  out= data.frame(mean_difference = obs,
                  n1= n1,
                  n2= n2,
                  p= ifelse(p==0,"p < 0.0002",p))
  out
}
```

```

}
# EXAMPLE----
# create random variables from non-normal skewed distributions
set.seed(1)
a=rlnorm(25, meanlog = 1, sdlog = 1)
b=rlnorm(25, meanlog = 1.5, sdlog = 2)
# plot
boxplot(a,b)
# parametric t-test (makes wrong assumptions about distribution)
t.test(a,b)
# nonparametric permutation test (high power)
permutation.test(a,b)
# nonparametric rank test (low power)
wilcox.test(a,b)
-----
Gerry Carter

```

Decision letter (RSOS-181942.R0)

31-Jan-2019

Dear Dr O'Mara

On behalf of the Editors, I am pleased to inform you that your Manuscript RSOS-181942 entitled "Common noctules exploit low levels of the aerosphere" has been accepted for publication in Royal Society Open Science subject to minor revision in accordance with the referee suggestions. Please find the referees' comments at the end of this email.

The reviewers and handling editors have recommended publication, but also suggest some minor revisions to your manuscript. Therefore, I invite you to respond to the comments and revise your manuscript.

- Ethics statement

- Data accessibility

<http://datadryad.org/submit?journalID=RSOS&manu=RSOS-181942>

- **Competing interests**

- **Authors' contributions**

- **Acknowledgements**

- **Funding statement**

Because the schedule for publication is very tight, it is a condition of publication that you submit the revised version of your manuscript before 09-Feb-2019. Please note that the revision deadline will expire at 00.00am on this date. If you do not think you will be able to meet this date please let me know immediately.

on behalf of Dr Safi Darden (Associate Editor) and Professor Kevin Padian (Subject Editor)
openscience@royalsociety.org

Reviewer comments to Author:

Reviewer: 1

Comments to the Author(s)

I salute you all on such an excellent contribution. The topic is so alluring and yet has received so little attention. Thank you for sorting this out.

I do, of course, have a couple of suggestions.

First, please put more information into the abstract ...give the numbers of bats tagged (8) and the time spent in contact with them this will put the size of the data base in clear context and it is appropriate to make it easier for the reader to get an idea of what is to come.

Second. Consider articulating the hypothesis that guided the work. I like the specific predictions, but if they arose from a guiding hypothesis, the impact of the work would be clearer to everyone. This will increase the impact and readership of the contribution.

Great work, lovely contribution

Brock Fenton

Reviewer: 2

Comments to the Author(s)

This paper describes the altitudinal movements of common noctule bats using pressure-sensitive radio-transmitters. The study is very well-written, and it provides novel and valuable information about how bats use the vertical dimension. The bats ascended briefly over 100 m in most foraging sessions, but they stayed >1 minute above that height in only 2 of 25 observed foraging sessions. The results made sense given what we know about how these bats forage. The authors did not detect that the bats increased their wingbeats during these ascents, but the bats did skip wingbeats and also glided. This information could be useful for informing policies for mitigating bat mortality at wind turbines. It is also terrific to see what animals are doing in places where we cannot normally observe them.

There were just a couple of places that I think the manuscript could be improved. First, it would be nice to get a bit more detail of the accuracy of the tags (given the possible change of air pressure caused by wingbeats). I suspect this information is somewhere and it would just be nice to provide it for the reader since so much depends on a rough idea of their accuracy. One way to do this (maybe) would be to put some standard-size error bars on the altitude data so one could infer what is a bat moving up and down versus what is simply error. Or if the tags are high-resolution enough, perhaps there would be no visible error bars?

Second, it would help to draw a better distinction between results of “no relationship” that the authors think actually means a relationship of zero (biologically irrelevant) versus when “no relationship” means something like “we detected no relationship probably due to a small effect size and a lot of noise”. My line comments for each section are below. For more powerful nonparametric tests, I provide some R code for an alternative to rank tests.

INTRO

First paragraph: Regarding the “aerosphere” niche, can you briefly clarify what you mean using some concrete examples? Is an insect flying one foot above the ground in “airspace” or are we only talking about very high up? (e.g. birds and bat feeding on migrating insects/birds/bats). What is the difference between airspace and aerosphere? How high does it need to be?

50-53 Sentence needs revision.

69 Define “higher conflict zones”

77 missing parenthesis

126 bats at our study site

129 A bit confusing: the first part is the hypothesis, and the second part is the prediction? Maybe reword as:

“If high-elevation ascents are a response to increased foraging effort, then ascents to higher altitudes should be preceded by increased variation in altitude changes caused by the bat searching for prey vertically.”

METHODS AND RESULTS

161 Can you briefly clarify how you measured the general foraging location? (e.g. using radio signal strength to get distance and direction?)

168 “The transmitters had been previously calibrated for accuracy using the altimeter of a small aircraft with $r^2 = 0.99$ [38].” Can you give more information here? What is the scale or unit of analysis for $r^2=0.99$? Given the wingbeats affect on air pressure, what is the error bars (roughly speaking) that are associated with altitude measures for a flying bat? For example, are these tags accurate to 0.1 meter, 1 meter, 10 meter, 40 meters?

197 Sample size is 4546 air pressure measures?

Figure 1. This plot is your method for counting wingbeats? If so, maybe say that in Figure legend.

219 “we assume that flights with large changes in height indicate greater foraging effort.” Does dropping in altitude also indicate foraging?

Figure 2. I think it would help to connect the dots with a line so we can see the variation over time (or the error) more clearly, and to expand the x-axis and stack the plots in 4 rows. Figure 2 might be a good place to illustrate how you interpret the data. How many ascents are counted in each panel?

Fig. 3A says “there are no differences” but the distributions looks different. By ranking the data, you cannot detect that the high values are much higher in the descending rates, because the exact values are lost. Data look lognormal, and might be better to visualize and analyze them after a log transformation like $\log(x+1)$.

A nonsignificant test doesn’t show that two variables are similar or drawn from the same distribution. So I think it’s better to say “no differences detected” rather than saying “the same”, “no effect”, “no differences” or “no relationship” when not detecting a difference with low power (unless the mean effect is precisely near zero). This also applied to lines 294, 302, or 305 and first paragraph of discussion.

Many readers do know when to translate “no effect” as “inconclusive”, but many do not. If you do think the effect is about zero, I think it makes sense to say “no effect” but then you should present a near-zero effect size and confidence interval that is close around zero.

Also, I think ranked data tests like Mann-Whitney U tests are just too low power to argue

convincingly for differences or lack of differences between two groups. Below is some R code for a more powerful non-parametric permutation test that can be used of Mann-Whitney instead to test differences between two means drawn from any distribution.

I think the Figure 4 caption (line 285 or 286) is missing some words (e.g. "are shown")

Besides my comment above about "no relationship" vs "not detecting a relationship" , the discussion reads well to me.

```
-----

# PERMUTATION TEST-----

permutation.test <- function(x,y){
  perms <- 5000 # get number of permutations
  n1 <- length(na.omit(x)) # get sample size of group 1
  n2 <- length(na.omit(y)) # get sample size of group 2
  obs <- mean(x, na.rm=T) - mean(y, na.rm=T) # get mean difference
  exp <- rep(NA, perms)
  for (i in 1:perms) {
    # shuffle data to random groups
    xy <- sample(c(x,y))
    x2 <- xy[1:n1]
    y2 <- xy[(n1+1):length(xy)]
    #get mean difference between random groups
    exp[i] <- mean(x2, na.rm=T) - mean(y2, na.rm=T)
  }
  # get two-sided p-value
  p <- mean(abs(obs)<=abs(exp))
  # get results
  out= data.frame(mean_difference = obs,
                  n1= n1,
                  n2= n2,
                  p= ifelse(p==0,"p < 0.0002",p))
  out
}

# EXAMPLE----
# create random variables from non-normal skewed distributions
set.seed(1)
a=rlnorm(25, meanlog = 1, sdlog = 1)
b=rlnorm(25, meanlog = 1.5, sdlog = 2)

# plot
boxplot(a,b)

# parametric t-test (makes wrong assumptions about distribution)
t.test(a,b)

# nonparametric permutation test (high power)
permutation.test(a,b)

# nonparametric rank test (low power)
```

wilcox.test(a,b)

Gerry Carter

Author's Response to Decision Letter for (RSOS-181942.R0)

See Appendix A.

Decision letter (RSOS-181942.R1)

06-Feb-2019

Dear Dr O'Mara,

I am pleased to inform you that your manuscript entitled "Common noctules exploit low levels of the aerosphere" is now accepted for publication in Royal Society Open Science.

on behalf of Dr Safi Darden (Associate Editor) and Professor Kevin Padian (Subject Editor)
openscience@royalsociety.org

Appendix A

Dear Editors and Drs. Fenton & Carter,

Thank you for the positive receipt of our manuscript and the valuable suggestions. We have responded below to each suggestion in our current revision. We particularly thank Dr. Carter for his suggestion on changing the non-parametric analyses and the included code.

Best,

Teague O'Mara

Reviewer comments to Author:

Reviewer: 1

Comments to the Author(s)

I salute you all on such an excellent contribution. The topic is so alluring and yet has received so little attention. Thank you for sorting this out.

I do, of course, have a couple of suggestions. First, please put more information into the abstract ...give the numbers of bats tagged (8) and the time spent in contact with them this will put the size of the data base in clear context and it is appropriate to make it easier for the reader to get an idea of what is to come.

>We have added this to the abstract.

Second. Consider articulating the hypothesis that guided the work. I like the specific predictions, but if they arose from a guiding hypothesis, the impact of the work would be clearer to everyone. This will increase the impact and readership of the contribution.

> We have reformulated the guiding hypothesis about how & why bats ascend to high elevations at L 137

Great work, lovely contribution

Brock Fenton

Reviewer: 2

Comments to the Author(s)

This paper describes the altitudinal movements of common noctule bats using pressure-sensitive radio-transmitters. The study is very well-written, and it provides novel and valuable information about how bats use the vertical dimension. The bats ascended briefly over 100 m in most foraging sessions, but they stayed >1 minute above that height in only 2 of 25 observed foraging sessions. The results made sense given what we know about how these bats forage. The authors did not detect that the bats increased their wingbeats during these ascents, but the bats did skip wingbeats and also glided. This information could be useful for informing policies for mitigating bat mortality at wind turbines. It is also terrific to see what animals are doing in places where we cannot normally observe them.

There were just a couple of places that I think the manuscript could be improved. First, it would be nice to get a bit more detail of the accuracy of the tags (given the possible change of air pressure caused by wingbeats). I suspect this information is somewhere and it would just be nice to provide it for the reader since so much depends on a rough idea of their accuracy. One way to do this (maybe) would be to put some standard-size error bars on the altitude data so one could infer what is a bat moving up and down versus what is simply error. Or if the tags are high-resolution enough, perhaps there would be no visible error bars?

Second, it would help to draw a better distinction between results of “no relationship” that the authors think actually means a relationship of zero (biologically irrelevant) versus when “no relationship” means something like “we detected no relationship probably due to a small effect size and a lot of noise”. My line comments for each section are below. For more powerful nonparametric tests, I provide some R code for an alternative to rank tests.

INTRO

First paragraph: Regarding the “aerosphere” niche, can you

briefly clarify what you mean using some concrete examples? Is an insect flying one foot above the ground in "airspace" or are we only talking about very high up? (e.g. birds and bat feeding on migrating insects/birds/bats). What is the difference between airspace and aerosphere? How high does it need to be?

> Aerosphere is the sum of the atmosphere in the planetary boundary layer and generally used to describe habitat. We use airspace interchangeably, and we have corrected this to be more consistent with the Kunz et al 2008 seminal description on the subject.

50-53 Sentence needs revision.

>revised

69 Define "higher conflict zones"

>modified to include turbine and the potential height range of the turbine blades

77 missing parenthesis

>check

126 bats at our study site

>check

129 A bit confusing: the first part is the hypothesis, and the second part is the prediction? Maybe reword as:

"If high-elevation ascents are a response to increased foraging effort, then ascents to higher altitudes should be preceded by increased variation in altitude changes caused by the bat searching for prey vertically."

>Thanks for the suggestion & we have reworded accordingly.

METHODS AND RESULTS

161 Can you briefly clarify how you measured the general foraging location? (e.g. using radio signal strength to get distance and direction?)

> included

168 "The transmitters had been previously calibrated for accuracy using the altimeter of a small aircraft with $r^2 = 0.99$

[38].” Can you give more information here? What is the scale or unit of analysis for $r^2=0.99$? Given the wingbeats affect on air pressure, what is the error bars (roughly speaking) that are associated with altitude measures for a flying bat? For example, are these tags accurate to 0.1 meter, 1 meter, 10 meter, 40 meters?

>This was an altitude-altitude calibration (L182). Tag sensitivity has been included at L177 and the error of each measurement is so small that visualizing it on the scales of our figures would not be productive.

Also, while it is possible that wing movements of the bat could impact air pressure, this is so small (bat wingbeats are relatively low power) that it can be ignored.

197 Sample size is 4546 air pressure measures?

>Corrected

Figure 1. This plot is your method for counting wingbeats? If so, maybe say that in Figure legend.

>Corrected

219 “we assume that flights with large changes in height indicate greater foraging effort.”

Does dropping in altitude also indicate foraging?

>We assume yes.

Figure 2. I think it would help to connect the dots with a line so we can see the variation over time (or the error) more clearly, and to expand the x-axis and stack the plots in 4 rows. Figure 2 might be a good place to illustrate how you interpret the data. How many ascents are counted in each panel?

> We have re-formatted to stack the panels vertically and provided a bit more interpretation in the text.

Fig. 3A says “there are no differences” but the distributions looks different. By ranking the data, you cannot detect that the high values are much higher in the descending rates, because the exact values are lost. Data look lognormal, and might be better to visualize and analyze them after a log

transformation like $\log(x+1)$.

A nonsignificant test doesn't show that two variables are similar or drawn from the same distribution. So I think it's better to say "no differences detected" rather than saying "the same", "no effect", "no differences" or "no relationship" when not detecting a difference with low power (unless the mean effect is precisely near zero). This also applied to lines 294, 302, or 305 and first paragraph of discussion.

Many readers do know when to translate "no effect" as "inconclusive", but many do not. If you do think the effect is about zero, I think it makes sense to say "no effect" but then you should present a near-zero effect size and confidence interval that is close around zero.

Also, I think ranked data tests like Mann-Whitney U tests are just too low power to argue convincingly for differences or lack of differences between two groups. Below is some R code for a more powerful non-parametric permutation test that can be used of Mann-Whitney instead to test differences between two means drawn from any distribution.

> We have rephrased accordingly & used Gerry's suggested permutation test – we agree about the low power, but as a first approximation it's not a bad start. Thanks for including the code! We've detected differences in means that were not immediately detectable before but are line with what one would predict.

I think the Figure 4 caption (line 285 or 286) is missing some words (e.g. "are shown")

> Corrected

Besides my comment above about "no relationship" vs "not detecting a relationship", the discussion reads well to me.

>Thanks

```
# PERMUTATION TEST-----
```

```
permutation.test <- function(x,y){  
  perms <- 5000 # get number of permutations  
  n1 <- length(na.omit(x)) # get sample size of group 1  
  n2 <- length(na.omit(y)) # get sample size of group 2  
  obs <- mean(x, na.rm=T) - mean(y, na.rm=T) # get mean  
  difference  
  exp <- rep(NA, perms)  
  for (i in 1:perms) {  
    # shuffle data to random groups  
    xy <- sample(c(x,y))  
    x2 <- xy[1:n1]  
    y2 <- xy[(n1+1):length(xy)]  
    #get mean difference between random groups  
    exp[i] <- mean(x2, na.rm=T) - mean(y2, na.rm=T)  
  }  
  # get two-sided p-value  
  p <- mean(abs(obs)<=abs(exp))  
  # get results  
  out= data.frame(mean_difference = obs,  
                  n1= n1,  
                  n2= n2,  
                  p= ifelse(p==0,"p < 0.0002",p))  
  out  
}
```

```
# EXAMPLE-----
```

```
# create random variables from non-normal skewed  
distributions
```

```
set.seed(1)
```

```
a=rlnorm(25, meanlog = 1, sdlog = 1)
```

```
b=rlnorm(25, meanlog = 1.5, sdlog = 2)
```

```
# plot
```

```
boxplot(a,b)
```

```
# parametric t-test (makes wrong assumptions about  
distribution)
```

```
t.test(a,b)
```

```
# nonparametric permutation test (high power)
permutation.test(a,b)
```

```
# nonparametric rank test (low power)
wilcox.test(a,b)
```

Gerry Carter